# Glassy PEEK-WC vs. Rubbery Pebax®1657 Polymers: Effect on the Gas Transport in CuNi-MOF Based Mixed Matrix Membranes

**Elisa Esposito** [1],*, **Rosaria Bruno** [2], **Marcello Monteleone** [1], **Alessio Fuoco** [1],
**Jesús Ferrando Soria** [3], **Emilio Pardo** [3], **Donatella Armentano** [2] and
**Johannes Carolus Jansen** [1],*

1   Institute on Membrane Technology, CNR-ITM, Via P. Bucci 17/C, 87036 Rende (CS), Italy;
    m.monteleone@itm.cnr.it (M.M.); a.fuoco@itm.cnr.it (A.F.)
2   Dipartimento di Chimica e Tecnologie Chimiche (CTC), Università della Calabria, 87036 Cosenza, Italy;
    rosaria.bruno@unical.it (R.B.); donatella.armentano@unical.it (D.A.)
3   Departamento de Química Inorgánica, Instituto de Ciencia Molecular (ICMol), Catedrático José Beltrán
    Martínez, 2, Universidad de Valencia, 46980 Paterna, Valencia, Spain; jesus.ferrando@uv.es (J.F.S.);
    emilio.pardo@uv.es (E.P.)
*   Correspondence: e.esposito@itm.cnr.it (E.E.); johannescarolus.jansen@cnr.it (J.C.J.);
    Tel.: +39-0984-492008 (E.E.); +39-0984-492031 (J.C.J.)

**Abstract:** Mixed matrix membranes (MMMs) are seen as promising candidates to overcome the fundamental limit of polymeric membranes, known as the so-called Robeson upper bound, which defines the best compromise between permeability and selectivity of neat polymeric membranes. To overcome this limit, the permeability of the filler particles in the MMM must be carefully matched with that of the polymer matrix. The present work shows that it is not sufficient to match only the permeability of the polymer and the dispersed phase, but that one should consider also the individual contributions of the diffusivity and the solubility of the gas in both components. Here we compare the gas transport performance of two different MMMs, containing the metal–organic framework CuNi-MOF in the rubbery Pebax®1657 and in the glassy poly(ether-ether-ketone) with cardo moiety, PEEK-WC. The chemical and structural properties of MMMs were investigated by means of FT-IR spectroscopy, scanning electron microscopy and EDX analysis. The influence of MOF on the mechanical and thermal properties of both polymers was investigated by tensile tests and differential scanning calorimetry, respectively. The MOF loading in Pebax®1657 increased the ideal $H_2/N_2$ selectivity from 6 to 8 thanks to an increased $H_2$ permeability. In general, the MOF had little effect on the Pebax®165 membranes because an increase in gas solubility was neutralized by an equivalent decrease in effective diffusivity. Instead, the addition of MOF to PEEK-WC increases the ideal $CO_2/CH_4$ selectivity from 30 to ~48 thanks to an increased $CO_2$ permeability (from 6 to 48 Barrer). The increase in $CO_2$ permeability and $CO_2/CH_4$ selectivity is maintained under mixed gas conditions.

**Keywords:** mixed matrix membrane; glassy polymer; rubbery polymer; PEEK-WC; Pebax®1657; gas separation; CuNi-MOF

---

## 1. Introduction

The research field on materials for gas separation membranes is constantly expanding due to the pressing industrial request for more performing materials to employ in gas treatment, such as hydrogen recovery ($H_2/CO_2$) [1], oxygen-enriched air production ($O_2/N_2$) [2,3], biogas up-grading [4,5] and

natural gas treatment ($CO_2/CH_4$) [6], and post-combustion carbon capture from flue gas ($CO_2/N_2$) [7]. This increasing interest is dictated by the advantages of membrane technology compared to traditional gas separation techniques. Gas separation by means of membrane technology is an economical process; it is easily scalable and it can be used in non-drastic temperature and pressure conditions, which are more environmentally friendly. Despite numerous efforts to develop new materials for gas separation, the membrane market still needs to overcome some challenges. In fact, the highly permeable rubbery polymers present low selectivity, and the highly-selective glassy polymers are less permeable. To overcome this trade-off between permeability and selectivity, extensively reported by Robeson in 1991 and in 2008 [8,9], researchers are focusing on the design and development of hybrid membranes based on the combination of two different materials in order to get the advantages of both [10]. According to this concept, a valid strategy is the embedding of porous metal-organic frameworks (MOFs) in the polymer matrix in order to obtain mixed matrix membranes (MMMs) with enhanced gas transport properties [11–16]. MOFs are an attractive new class of microporous materials built by the combination of metal atoms/clusters with a wide variety of organic ligands, which can be specifically designed for improving the compatibility with the organic polymer phase and with high specificity for gases [17,18]. The gas transport in polymer membranes occurs following the solution-diffusion mechanism in which permeability and selectivity are determined by kinetic parameters (diffusion) and thermodynamic factors (solubility), via $P = D * S$; $\alpha_{\frac{x}{y}} = \frac{D_x}{D_y} * \frac{S_x}{S_y}$. The addition of MOFs can improve the gas transport properties of the neat polymer by influencing these two parameters [10]. The diffusion, being a kinetic phenomenon, is strongly linked to the free volume of the polymer materials and to the molecular size of the penetrating species. In the MOFs, the link between metals and organic units forms a three-dimensional structure with cages having well-defined shape and size, which can improve free volume elements of a matrix, increasing gas transport in terms of permeability. At the same time, the cages of MOF forming a preferential pathway for a specific gas can act as a molecular sieve, increasing membrane selectivity. On the contrary, the main factor that determines the solubility in a polymer matrix is the ability of the penetrant gases to condense, which in turn is correlated with the interactions that occur between the gas and the matrix of membrane. MOFs chemical structure can be easily designed or functionalized synthetically in order to improve its affinity for specific gases [19]. In this case, the membrane permeability is expected to improve as a consequence of an increased contribution in solubility due to the gas condensability in the MOF. Furthermore, the introduction of chemical groups with a specific affinity to one gas in a mixture will also increase its selectivity. The choice of the materials combination for MMMs preparation must be made on the basis of specific physical and chemical properties, in order to tailor in advance the membrane for a desired gas separation. For this reason, it is necessary to understand how the addition of MOF can influence the gas transport properties of different polymers, rubbery or glassy.

In this work, the gas transport properties for MMMs prepared by embedding the same oxamate-based MOF in two different polymers, Pebax®1657 rubbery polymer and PEEK-WC glassy polymer, will therefore be investigated. The oxamate-based MOF, previously reported [20], with formula $NiII_2\{NiII_4[CuII_2(Me_3mpba)_2]_3\}\cdot54H_2O$ (where $Me_3mpba$ is the N,N'-2,4,6-trimethyl-1,3-phenylenebis (oxamate) ligand), has already shown interesting gas separation properties [21].

Single-crystal X-ray diffraction measurements unveil the crystal structure of CuNi-MOF. The anionic $Ni^{II}_6Cu^{II}_6$ open-framework structure exhibits a pillared square/octagonal layer architecture, where nickel(II) and copper(II) ions are located on the vertices and midpoints of the edges, respectively, featuring three types of pores, different in size and shape, propagating along the c axis and enfolding up to 60% of the total lattice volume. Free nickel(II) cations are further accommodated within pores of the MOF. It consists of regularly spaced, small, almost square-sized pores (with a virtual diameter of ca 0.4 nm) and two kinds of hydrophobic and hydrophilic octagonal pores (Figure 1) resulting from the different disposition of the trimethyl-substituted phenylene spacers, pointing inwards or outwards from the voids, which accounts for their virtual diameters of ca. 1.5 and 2.2 nm, respectively (Figure 1). PEEK-WC is a glassy poly(ether-ether-ketone) with a cardo group in the polymer backbone,

which makes it soluble in various common organic solvents. It has been used extensively in membrane preparation and characterization under the names PEEK-WC, PEEKWC or PEK-C [22]. Membranes have been prepared in the form of hollow fibres for liquid filtration [23] and as flat films for liquid filtrations [24] or for gas separation [25–27] and numerous other applications [22]. In this work, we take advantage of its good solubility in chloroform for the preparation of mixed matrix membranes by the solvent evaporation method. PEEK-WC presents a high selectivity but a too low permeability to be interesting as a material for industrial application. For this reason, different researchers have tried to improve its permeability without loss in selectivity by means of nanoparticle addition. An extensive protocol to obtain MMMs based on PEEK-WC, embedding NaA (LTA) zeolite, was developed in order to improve gas separation parameters [28]. However, no one procedure has enhanced gas transport properties of the neat polymer, and a decrease in selectivity was observed probably due to a presence of defects at interface between polymer and zeolite. Some methods have been developed in order to further improve the adhesion of PEEK-WC for 3D-mesoporous nanoparticles. For example, the MIL-101 was functionalized with a sulfonic acid group to further increase the affinity for the polymer matrix. The resulting membranes have shown an improved $CO_2$ permeability and $CO_2$/gas selectivity. The increased selectivity was mainly attributed to the increased polar interaction between $CO_2$ and sulfonic acid groups, as well as the good filler–polymer interface compatibility [29].

**Polymer Structure.**

**CuNi-MOF Structure**

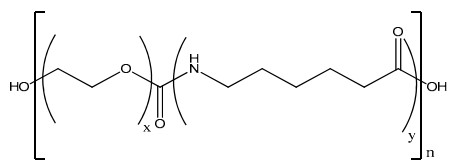

**Pebax®1657**, x = 0.6, y = 0.4, density = 1.14 g cm$^{-3}$

(**a**)

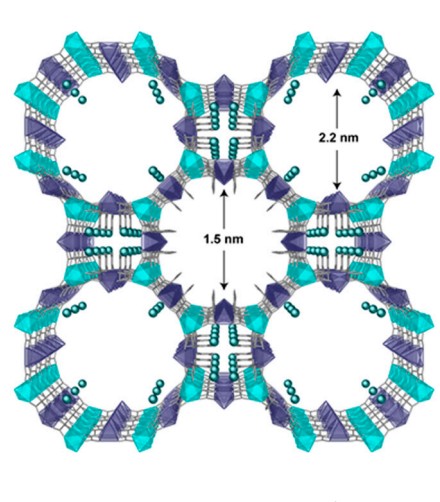

**PEEK-WC,** density = 1.25 g cm$^{-3}$

(**b**)

Density = 0.54 g cm$^{-3}$

(**c**)

**Figure 1.** Chemical structure and density of (**a**) Pebax®1657, (**b**) PEEK-WC; and (**c**) CuNi-MOF. Cu and Ni atoms from the network are represented by cyan and green polyhedra, respectively, whereas organic ligands are depicted as sticks. Green spheres represent Ni$^{2+}$ metal ions in pores.

Pebax®1657 is a thermoplastic elastomer multiblock-copolymer containing linear chains of rigid polyamide segments interspaced with flexible polyether segments. The hard polyamide (PA) blocks supply mechanical strength and the presence of the polar ethylene oxide (PEO) group, increases the affinity for $CO_2$, allowing for a good $CO_2$/non-polar ($H_2$, $N_2$, and $CH_4$) species separation and permselectivity. Several membranes have been prepared in the form of composite hollow fibre membranes [30–32], hybrid and mixed matrix membranes for gas separation [33–43]. Mixed matrix membranes based on amine-functionalized and pristine MIL-53(Al) in Pebax1657 have shown an increased $CO_2$ separation and the higher permeability (PCO$_2$ 100 Barrer and $\alpha CO_2$/CH$_4$ = 17; $CO_2$/N$_2$ = 50), was attributed to high porosity introduced by the presence of the MOF, as well as the selective adsorption of $CO_2$ inside the MOF [41]. Liu et al. designed nanosheet-enhanced Pebax-MoS2

mixed matrix membranes for $CO_2$ capture, increasing $CO_2/N_2$ selectivity until 90 [44]. Composite hollow fibre membranes with a selective layer of Pebax®1657 and different functionalized Uio-66 MOF have demonstrated a simultaneous improvement in gas permeance and selectivity. The enhanced selectivity was attributed to an increased rigidity of the polymer matrix due to a formation of a hydrogen bond between the MOF and Pebax®1657 polymer chains [45]. Pebax®1657 containing $Cu_3BTC_2$-MOF have shown an increased $CO_2/CH_4$ ideal selectivity about 15% compared to the neat Pebax®1657 [46]. The improvement in selectivity was attributed to the improvement in the $CO_2$ solubility, which, having a strong quadrupole moment, presents a higher affinity with unsaturated Cu sites than $CH_4$, leading to a higher $CO_2$ permeability. In this work, the CuNi-MOF was dispersed in the Pebax®1657 rubbery polymer and in the PEEK-WC glassy polymer, which, presenting a Cu site and well-defined cages (1.5 nm and 2.2 nm), is expected to influence at the same time the solubility and diffusion coefficient. The aim is to understand the main factors that determine the behaviour when a MOF is dispersed within a glassy or rubbery polymeric matrix. FT-IR, EDX, DSC and SEM analyses investigate the potential chemical interaction between CuNi-MOF and polymers. Besides, pure gas permeability tests with six different gases were carried out to obtain a general understanding of the MMM performance, while permeation tests were also carried out with binary $CO_2/N_2$ and $CH_4/CO_2$ gas mixtures to evaluate its potential performance in a real separation process. These two gas pairs simulate flue gas in view of the potential use of the membranes in $CO_2$ capture, and biogas in view of the potential exploitation of renewable energy in a strongly emerging market.

## 2. Materials and Methods

Pebax®1657, a poly(ethylene-oxide) (PEO) and poly[imino(1-oxohexamethylene)] (PA6) multi-block co-polymer in the molar ratio 60/40, was kindly provided by Arkema in the form of pellets.

PEEK-WC was provided as a powder by the Institute of Applied Chemistry, Changchun, China, and was used as received, without further purification.

The CuNi-MOF was obtained as crystalline phase through a double cation-exchange reaction in the solid state by immersing crystals of $Mg^{II}_2\{Mg^{II}_4[Cu^{II}_2\text{-}(Me_3mpba)_2]_3\}\cdot45H_2O$ in saturated aqueous solutions of $Ni(NO_3)_2\cdot6H_2O$ for several weeks [47]. Alternatively, a large scale synthesis of $Ni_2^{II}\{Ni^{II}_4[Cu^{II}_2(Me_3mpba)_2]_3\}\cdot54H_2O$ (2) was carried out by direct reaction of two aqueous solutions of $Na_4[Cu_2(Me_3mpba)_2]\cdot4H_2O$ (0.1 mol) and $Ni(NO_3)_2\cdot6H_2O$ (0.13 mol) in water and subsequent addition, after filtration and re–suspension in water of the resulting compound, of 0.067 mol of $Ni(NO_3)_2\cdot6H_2O$ (Yield 99 %) [20].

### 2.1. Membranes Preparation

Mixed matrix membranes were prepared by loading different concentrations of CuNi-MOF (9 wt %, 17 wt %, 23 wt %) in Pebax®1657 and in PEEK-WC. The Pebax®1657 solution at 10 wt % was prepared by swelling polymer pellets in a mixture of distilled water and ethanol (ratio 30:70 wt/wt) at room temperature overnight, in agreement with previously-reported procedure [30]. Then a homogeneous dope solution was obtained by heating to 80 °C under magnetic stirring for at least 10 min. The CuNi-MOFs were dispersed and sonicated in water-ethanol mixture for 30 min and subsequently added to the Pebax®1657 solution under magnetic stirring. While, homogenous solution at 5 wt % of PEEK-WC was obtained by dissolving the polymer powder in chloroform at room temperature for 24 h. Concurrently, the CuNi-MOFs were separately dispersed in chloroform and sonicated in an ultrasonic bath for 30 min before filtering the PEEK-WC solution through glass wool into the suspension. One hour of mechanical stirring and 30 min of sonication made each solution homogeneous. The resulting solutions were cast into levelled flat Teflon petri-dish and left to evaporate over 2–3 days. The MMMs produced were dried in the oven at 70 °C under vacuum conditions for 24 h.

## 2.2. Membranes Characterization

### 2.2.1. Structural Characterization

Chemical and morphological analysis of membranes were performed by scanning electron microscopy (Phenom Pro X desktop SEM, Phenom-World). The samples for cross-section SEM characterization were prepared by freeze-fracturing in liquid nitrogen. Samples were analyzed without sputter-coating with gold. Elemental analysis was performed with the Phenom- Pro X desktop SEM, which is equipped with an energy dispersive X-ray spectroscopy detector (EDX). Infrared spectroscopy (FT-IR) analyses were performed by means of Spectrum Spotlight Chemical Imaging Instrument (PerkinElmer) in ATR mode.

### 2.2.2. Thermal and Mechanical Characterization

DSC analysis was carried out using a Pyris Diamond Differential Scanning Calorimeter (Perkin Elmer, Shelton, CT, USA) equipped with an intracooler refrigeration system. Samples of 15–20 mg were used. Unless specified otherwise, three cycles were performed. The PEEK-WC samples were first heated from $-60\,°C$ to $300\,°C$, kept at this temperature for 1 min, and cooled down to $0\,°C$, where they were kept for 5 min. In the second run, the samples were heated up again to $300\,°C$. The DSC runs were performed at a scan rate of $15\,°C\,min^{-1}$. Before the measurements, the samples were kept at $50\,°C$ under vacuum for one night in order to remove possibly adsorbed water. The Pebax®1657-based samples were tested using an identical method in the range from $-40\,°C \rightarrow 220\,°C \rightarrow -75\,°C \rightarrow 225\,°C$.

Tensile tests were carried out at room temperature on a single column Universal Testing Machine, Zwick/Roell model Z2.5, equipped with a 50 N load cell and pneumatic clamps. The surface of one flat clamp was covered with adhesive rubber to avoid slipping or damaging of the samples, while the second clamp had a convex surface to increase the local pressure and to avoid the extraction of the sample. The average value and the standard deviation of the Young's modulus, the tensile strength and the maximum deformation were determined on a series of at least four samples. The sample width was 5 mm, and the grip-to-grip distances 40 and 30 mm for Pebax®16 and PEEK-WC, respectively. The tests were carried out at a deformation rate $20\,mm\,min^{-1}$ ($200\%\,min^{-1}$) for Pebax®1657-based samples and $6\,mm\,min^{-1}$ ($20\%\,min^{-1}$) for PEEK-WC-based samples.

### 2.2.3. Single Gas Permeation Method

Single gas permeation tests were carried out at $25\,°C$ and at a feed pressure of 1 bar, using a fixed-volume pressure increase instrument (ESSR), described elsewhere [48]. Permeability coefficients, $P$, and diffusion coefficients, $D$, were determined by the time-lag method [49]. The permeability coefficient, $P$, is calculated from the permeation curve in the steady state described by:

$$P_t = P_0 + (dp/dt)_0 \cdot t + \frac{RT \cdot A}{V_p \cdot V_m} \cdot \frac{p_f \cdot p}{l}\left(t - \frac{l^2}{6D}\right) \tag{1}$$

The last term in (Equation (1)) represents the so-called permeation time lag, $\Theta$, which is inversely proportional to the diffusion coefficient of the gas:

$$\Theta = \frac{l^2}{6D} \tag{2}$$

The approximate gas solubility coefficient, $S$, was obtained indirectly as the ratio of the permeability to the diffusion coefficient by assuming the solution-diffusion transport mechanism:

$$S = P/D \tag{3}$$

Permeabilities are reported in Barrer [1 Barrer = $10^{-10}$ cm$^3$ (STP) cm cm$^{-2}$ s$^{-1}$ cm Hg$^{-1}$].

### 2.2.4. Mixed Gas Permeation Measurement

Mixed gas permeation tests were carried out using a custom made constant pressure/variable volume instrument, described in detail elsewhere [50], equipped with a quadrupole mass filter (HPR-20 QIC Benchtop residual gas analysis system, Hiden Analytical, Warrington, UK). Measurements were performed from 1 to 6 bar(a) with two binary gas mixtures of $CO_2/N_2$ (15:85 vol %) and $CO_2/CH_4$ (35:65 vol %), simulating flue gas and biogas, respectively. Argon was used as a sweeping gas and as the internal standard for the calculation of the permeate gas composition.

## 3. Results and Discussion

### 3.1. Chemical and Morphological Characterization

The MOF loading in polymer matrix was evidenced by the distinctive color of obtained membranes, and it was accurately confirmed by the EDX analysis, which revealed the attendance of Cu and Ni metals in both Pebax®1657 and PEEK-WC polymers (See SI, Figure S1a,b, respectively). The chemical characterization of neat membranes and MMMs, as well as the interaction between polymer chains and nanoparticles were investigated by FTIR-ATR. Figure 2b,d shows the spectrum for neat Pebax®1657 and for Pebax®1657/CuNi-MOF membranes. The functional groups of Pebax®1657 give characteristic peaks at 3297 cm$^{-1}$ for N-H group, at 2943 and 2859 cm$^{-1}$ attributed to the asymmetric and symmetric stretching of the C-H bound, 1731 cm$^{-1}$ for the symmetric stretches of carboxylate R-O-C=O, while asymmetric R-O-C=O at ca. 1430, 1636 cm$^{-1}$ for H-N–C=O and 1099 cm$^{-1}$ for –C–O–C.

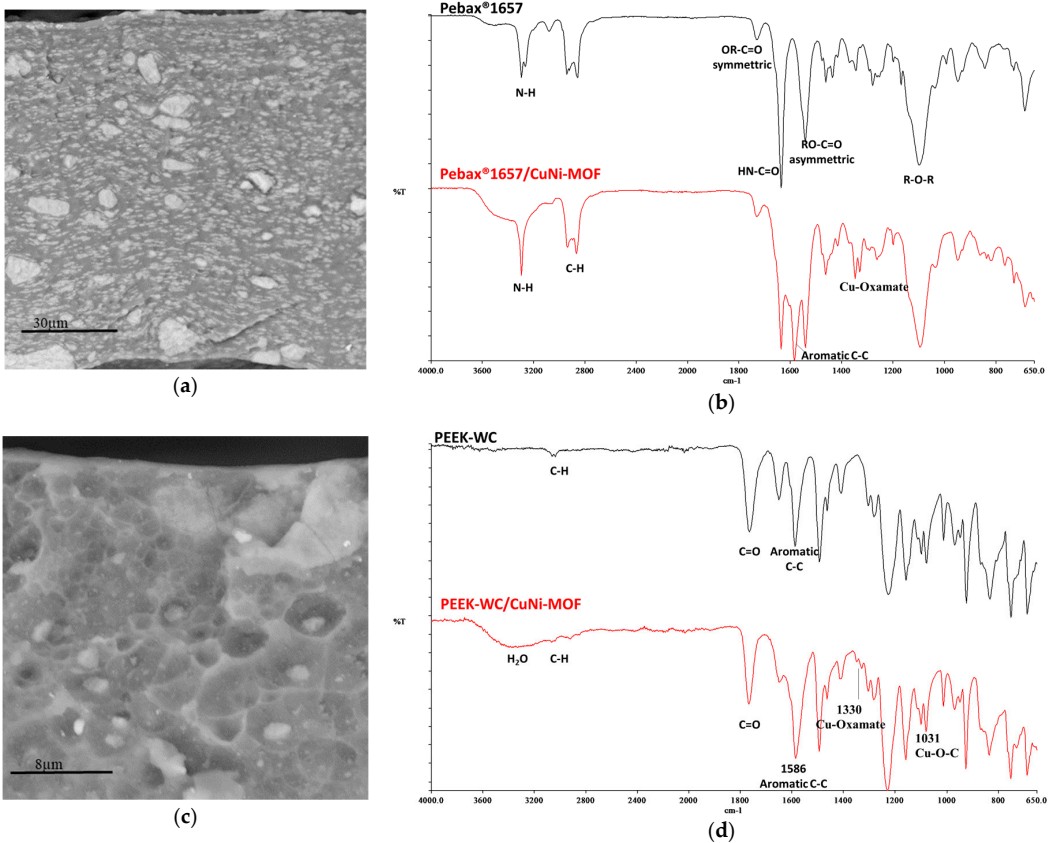

**Figure 2.** SEM image of the samples showing the interface between MOF and polymer phase (**a**) for Pebax®1657 and (**c**) PEEK-WC. FT-IR of neat polymer and MMMs with 23 wt % of CuNi, for Pebax®1657 (**b**) and PEEK-WC (**d**).

In the spectrum of Pebax®1657/CuNi-MOF, characteristic peaks appear at 1578 cm$^{-1}$ due to C–C aromatic band of MOF's phenylene, at 1608 cm$^{-1}$ of stretching of oxamate ligand of MOF and a peak at 1330 cm$^{-1}$, which could be considered diagnostic of the presence of bridging oxamate between Cu and Ni [51]. Finally, the characteristic peaks between 2900 cm$^{-1}$ and 2800 cm$^{-1}$ due to the symmetric and asymmetric C-H stretching are slightly shifted to lower wavelengths, confirming the interaction of hydrogen bonds at the interface between MOF and Pebax®1657, as it was also seen by Khosravi et al. for Pebax1657/CuBTC mixed matrix membranes [52]. Figure 2d offers a comparison of PEEK-WC membrane and PEEK-WC/CuNi spectrum. The functional groups of PEEK-WC give characteristic peaks at 3052 cm$^{-1}$ and 3075 cm$^{-1}$ for aromatic C–H stretch, at 1767 cm$^{-1}$ for ketonic and esteric C=O stretching, and at 1589 cm$^{-1}$ for the C–C aromatic band [53]. In PEEK-WC/CuNi, the peak for the C-C aromatic band is shifted on the lower wavelengths, probably due to the $\pi \cdots \pi$ stacking interaction between the benzene rings of polymers and MOF. In fact, when the ring is very conjugated, a weak band can be observed at around 1580 cm$^{-1}$, such as the one is visible from the PEEK-WC/CuNi spectrum. The peaks due to the stretch of the aromatic C–H appear to be totally fused with the water peak (3370 cm$^{-1}$) that reveals the relatively hydroscopic nature of the MOFs. The interaction between the Pebax®1657 and CuNi-MOF was also confirmed by SEM analysis, and in the enlarged particular of Figure 2a, it is possible to see that polymer phase of Pebax®1657creates a sort of circular compressed polymer region around the MOF, which Koros defined as "Case I matrix rigidification" [37]. On the other hand, the distribution of MOF in the PEEK-WC polymer produces a defined and regular network structure (Figure 2c) with a homogenous dispersion of nanoparticles without significant sedimentation across the membranes (See SI Figure S-2). Instead, the addition of MOFs in Pebax®1657 shows no significant agglomeration at the lowest 9 wt % that becomes more significant at highest 29 wt % concentration.

### 3.2. Mechanical and Thermal Properties

All membranes are dense mixed matrix membranes with good mechanical resistance for handling. The results of the tensile tests, performed on the nanocomposite films obtained on the series with different concentrations of CuNi-MOFs in Pebax®1657 and in PEEK-WC, are shown in Figure 3. The PEEK-WC presents a Young's module about 1.14 GPa [54] and Pebax®1657 has a Young's module about 0.10 GPa [55]. In both polymers, the loading of CuNi-MOF increases the Young's modulus. The increase of the Young's modulus indicates an increase of stiffness for both polymers, increasing the MOF concentrations. The glassy PEEK-WC presents a higher modulus compared to the rubbery Pebax®1657, as expected, and it remains higher even when the MOF concentration was increased. On the other hand, the maximum deformation is higher in the more flexible MMMs based on the rubbery Pebax®1657 compared to the MMMs based on the rigid glassy PEEK-WC.

The break strength decreases for Pebax®1657/CuNi membranes, indicating that the membranes become weaker compared to the neat polymer. On the contrary, the break strength increases for PEEK-WC/CuNi membranes, suggesting a good adhesion between MOF and polymer that makes the MMMs stronger than the neat PEEK-WC membrane. The thermal properties of the membranes were studied using DSC analysis, and the results are displayed in Figure 4. For neat Pebax®1657, the two dominant endothermal peaks at 19 °C and 200 °C are attributed to the fusion of the crystalline fraction of the soft poly(ethylene oxide) (PEO) blocks and the hard polyamide (PA) blocks, respectively. In the presence of MOF, the melting enthalpy of both PEO and PA decreases. However, the peak maximum shifts to a higher temperature, indicating that the MOF stabilizes the crystalline PEO and that it mainly interacts with this phase. On the other hand, no significant changes occur in the glass transition temperature of PEEK-WC at 230 °C as a function of the MOF concentration.

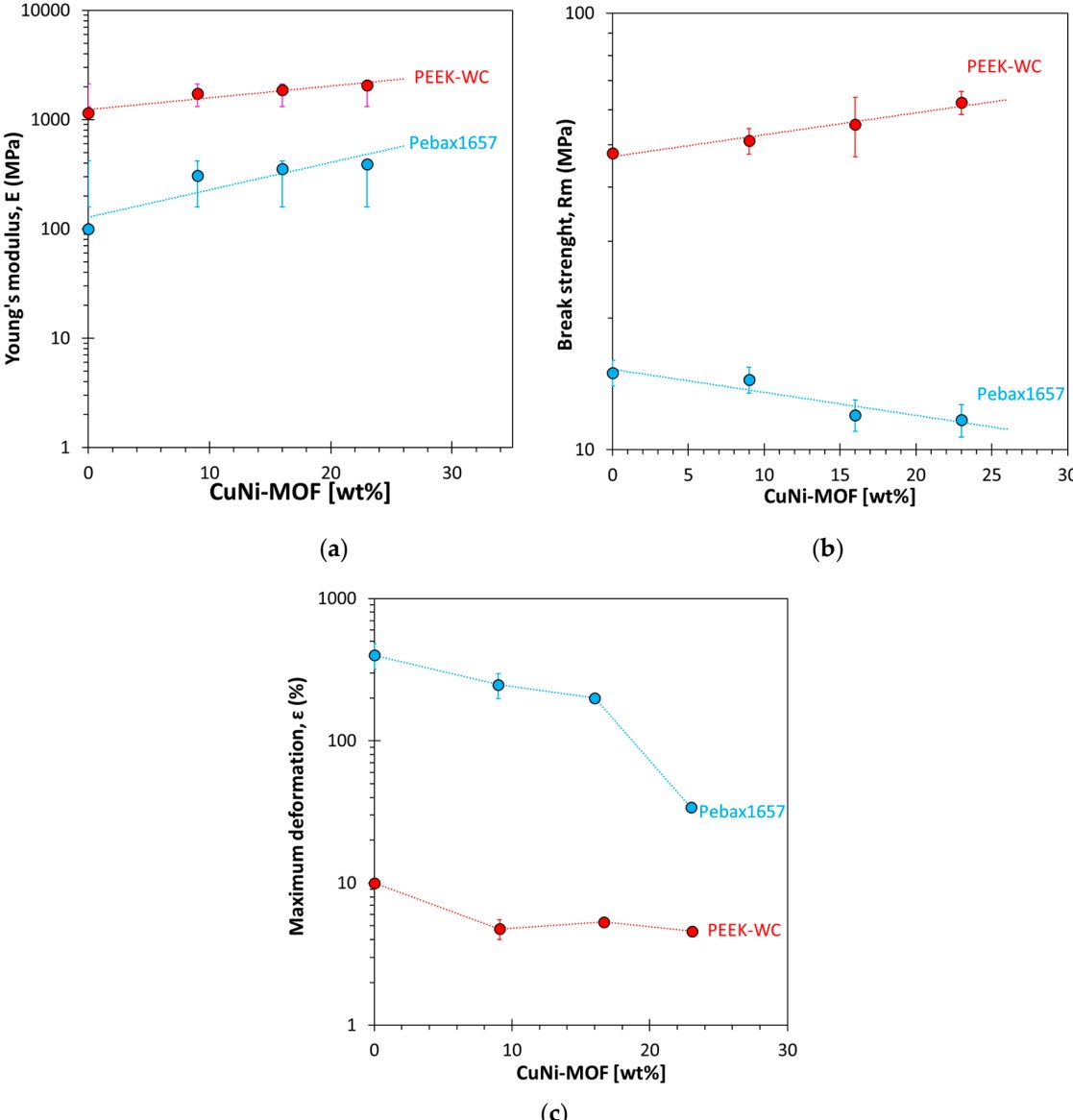

**Figure 3.** Young's modulus (**a**), tensile strength (**b**) and maximum deformation (**c**) as a function of the CuNi-MOF concentration in Pebax®1657 and PEEK-WC. Values of the neat PEEK-WC [50] and Pebax [51] samples from the literature. Trend lines are given as a guide to the eye.

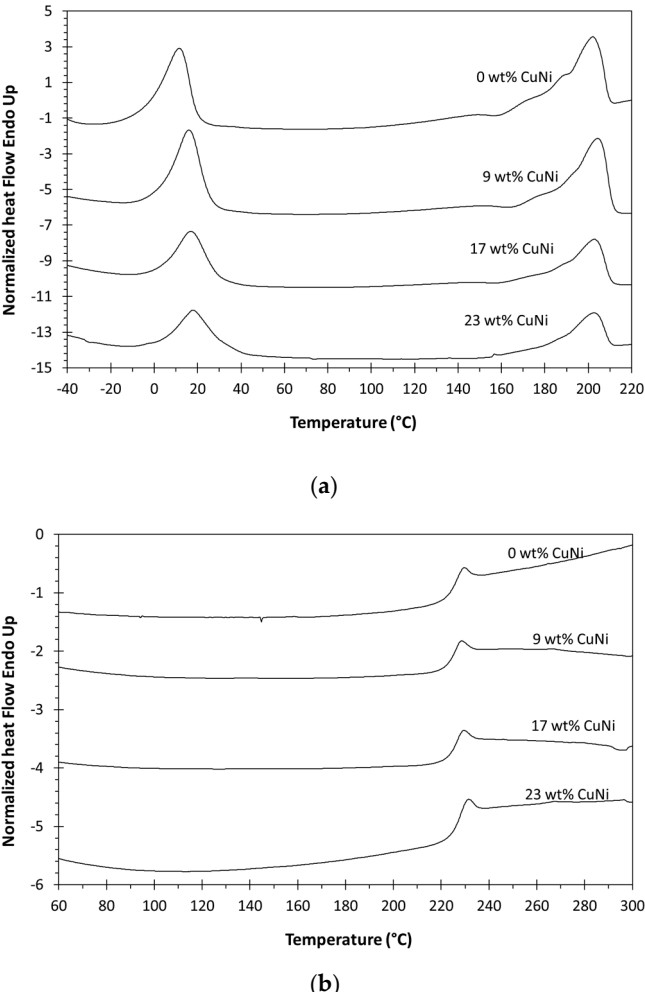

**Figure 4.** DSC thermograms for the membrane of (**a**) neat Pebax1657 and (**b**) neat PEEK-WC and MMMs with different concentration of CuNi-MOF during the second eating.

### 3.3. Pure Gas Transport Properties

Single gas permeation experiments were carried out in the order $H_2$, $O_2$, $N_2$, $CH_4$ and $CO_2$ at 25 °C, but repeated experiments with $N_2$ and $O_2$ at the end of the cycle were identical and revealed no structural changes in the material. An overview of the results of the Pebax®1657-based MMMs and PEEK-WC based MMMs with different filler loadings is given in Figures 5 and 6, respectively. All permeability, diffusivity and solubility data, and their respective selectivities, are given in the supporting information (See SI Tables S2 and S3). The dispersion of CuNi-MOF does not produce a substantial change in the permeability of the rubbery Pebax®1657. The most permeable species is $CO_2$, confirming a solubility controlled transport ($CO_2 > H_2 > CH_4 > O_2 > N_2$), typical for the rubbers (Figure 5a). A remarkably strong decrease in the effective diffusion coefficients of all gases as a function of the MOF loading (Figure 5b) is balanced by a proportional increase in their solubility coefficients (Figure 5c). This results in a similar permeability and selectivity (Figure 5d) for MMMs as in the neat Pebax®1657, with the only exception of the $H_2/CH_4$ gas pair, which suggests a favourable effect of the MOF on the size-selectivity. This is confirmed by the slight increase in the diffusion selectivities (Figure 5e), apparently because the MOF's provide preferential diffusion paths for smaller gases.

While the gas solubility upon addition of the MOF to Pebax increases as expected, given the generally high sorption capacity of MOFs for gases, it was not expected that this increase was more or less similar for all gases, and it was even less expected that the porous fillers would effectively lead to a decrease in the diffusion coefficient. The explanation is that the filler does not effectively decrease

the diffusion coefficient in the membrane, but the highly-sorbing MOF transforms the permeation experiment in a sort of breakthrough experiment. The reason for the apparently slower diffusion is the accumulation of the gas inside the pores of the filler, similar to the phenomenon of immobilizing adsorption [56,57], but in this case it has a virtually negligible net effect on the permeability.

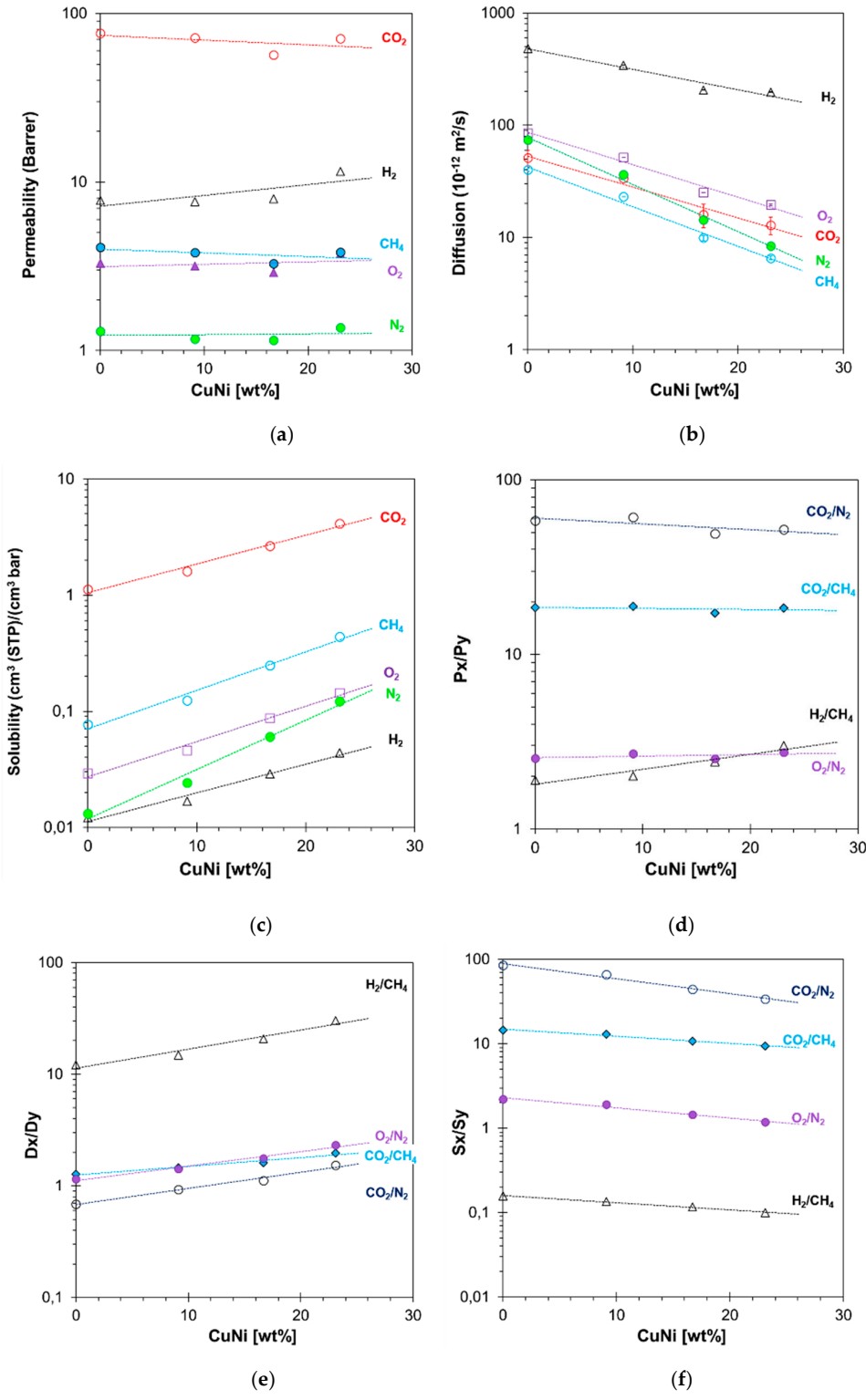

**Figure 5.** (**a**) Permeability, (**b**) diffusivity, and (**c**) solubility coefficients, and their respective selectivity towards $N_2$ (**d,e,f**) for each gas as functions of the weight percentage of CuNi-MOF in Pebax®1657.

In contrast to the effect in Pebax®1657, CuNi-MOF causes a drastic increase in the gas permeability for all gases as function of MOFs loading in the glassy PEEK-WC, and the order of permeation obeys that of the diffusion-controlled gas transport mechanism with $H_2 > CO_2 > O_2 > N_2 > CH_4$, typical for glassy polymers (Figure 6a). The increase in permeability is a direct consequence of an increase in diffusion coefficient (Figure 6b), whereas there is no substantial change in solubility coefficient (Figure 6c). The enhanced diffusion clearly indicates transport within the pore structure of CuNi-MOFs, which increases the total free volume of MMMs promoting the gas diffusion for all gases. The presence of MOF with chemical groups having a high affinity for $CO_2$ increases its solubility, its $CO_2/CH_4$ and $CO_2/N_2$ solubility selectivity (Figure 6f), and ideal selectivity for these two gas pairs (Figure 6d). Instead, the diffusion selectivity is not strongly affected by CuNi-MOF.

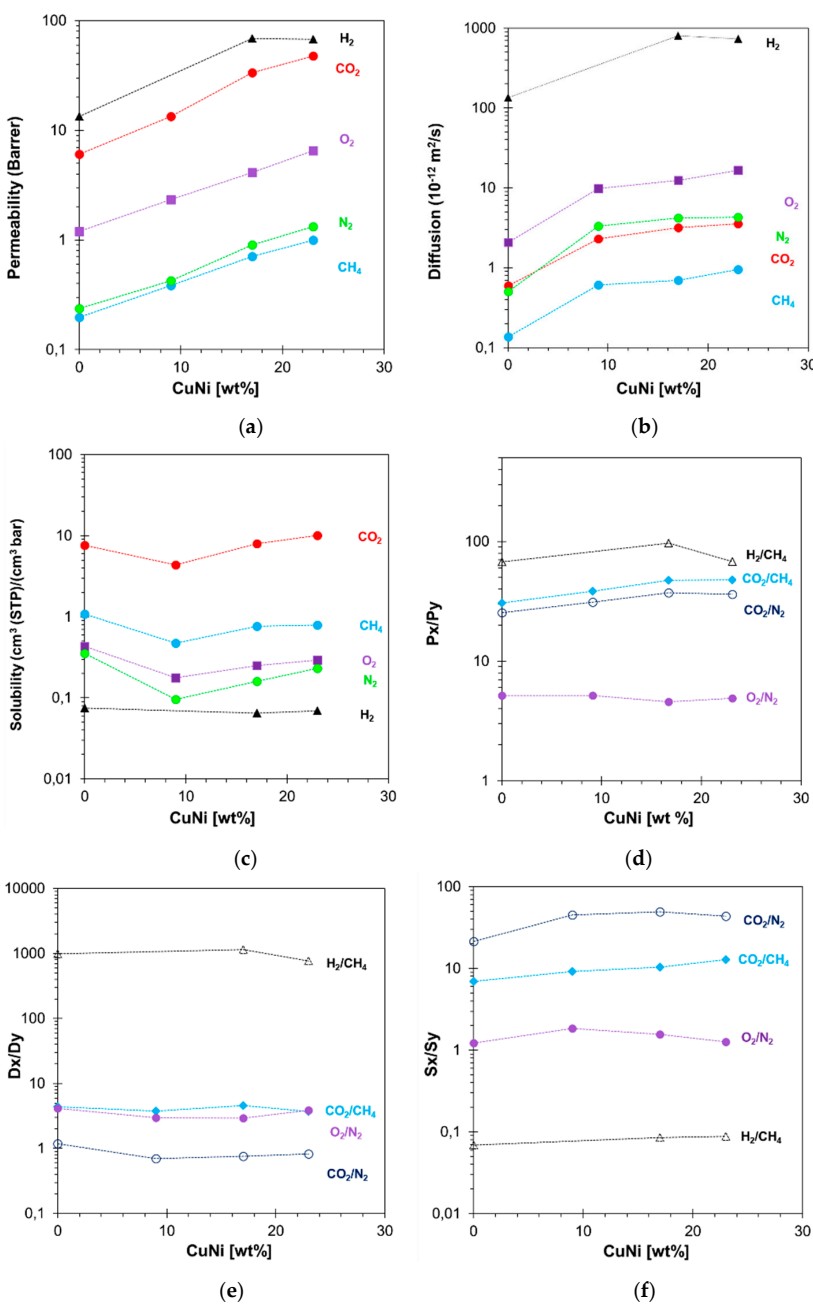

**Figure 6.** (**a**) Permeability, (**b**) diffusivity, (**c**) solubility coefficients, and respective $N_2$ selectivity (**d,e,f**) for each gas as functions of weight percentage of MOF loaded in PEEK-WC.

For a better understanding of the transport phenomena of gas separation membranes, the diffusion and solubility coefficients are often correlated with the molecular properties of gases [58]. Both neat polymers show a linear dependency of the diffusivity ($D$) on the squared gas diameter ($d^2_{eff}$) (Figure 7a,b), suggesting that the gas transport in neat polymers follows the diffusion solution model. In the presence of the MOF, both polymers show non-linearity of the $D$ vs. $d^2_{eff}$ correlation (Figure 7), which becomes more evident at higher MOF loading. A similar trend was recently observed for PIMs [59], where it was attributed to a different size-selectivity for small and large gas species due to the highly-interconnected free volume. These results suggest that the CuNi-MOF particles, with their internal void structure, have a similar effect on the gas diffusion of Pebax®1657 and PEEK-WC, introducing preferential diffusion pathways into the system which change the main transport mechanism, especially for the small molecules. As described above, the remarkably lower effective diffusion in Pebax®1657 is due to the much higher sorption capacity of the MOFs than of the neat Pebax®1657 [43]. Only for the $CO_2/N_2$ gas pair, there is a remarkable inversion from preferential $N_2$ diffusion in the neat Pebax®1657 to preferential $CO_2$ diffusion for the sample with the highest MOF concentration. Instead, in the glassy PEEK-WC, with much lower intrinsic diffusion coefficients in the neat polymer, the MOF has a strong positive effect on the total permeability of the mixed matrix membrane due to the far more rapid diffusion through the cages of the MOFs. In PEEK-WC, after an initial decrease, compared to the neat polymer, the solubility of all gases increases with the CuNi-MOF concentration. This is similar to what was observed in Pebax®1657, but only in PEEK-WC leads to an increasing permeability due to the positive effect on the diffusivity as well.

### 3.4. Mixed Gas Transport Properties

Mixed gas permeation measurements were carried out on the representative membranes with the highest MOF concentration in each polymer in view of two relevant industrial gas separations, namely biogas upgrading and $CO_2$ capture from flue gas, involving $CO_2$ separation from $CH_4$ and $N_2$, respectively (Figure 8) (See Table SI 4-5). For this purpose, measurements were performed from 1 to 6 bar(a) with simulated flue gas ($CO_2/N_2$, 15/85 vol %) and simulated biogas ($CO_2/CH_4$, 35/65 vol %). The glassy PEEK-WC MMM exhibits typical dual-mode behavior with the $CO_2/CH_4$ mixture, showing a decrease of $CO_2$ permeability as a function of the increased feed pressure, which causes a simultaneous and slightly smaller decrease of $CO_2/CH_4$ selectivity. This indicates that the free volume in the PEEK-WC phase and in the MOFs' pores is gradually becoming occupied by $CO_2$, and as a result, the $CH_4$ permeability slightly decreases as a function of pressure. Moreover, some hysteresis of $CO_2$ and $CH_4$ permeability is observed when reducing the pressure after the initial pressure increase steps. This is ascribed to a slight $CO_2$-induced dilation of the polymer, leaving a higher free volume when the pressure is decreased, and thus a higher $CH_4$ permeability and a slightly lower selectivity. In the separation of the $CO_2/N_2$ mixture, no further hysteresis takes place because of the lower $CO_2$ partial pressure, and for the same reason the $CO_2$ permeability is slightly higher than in the biogas mixture.

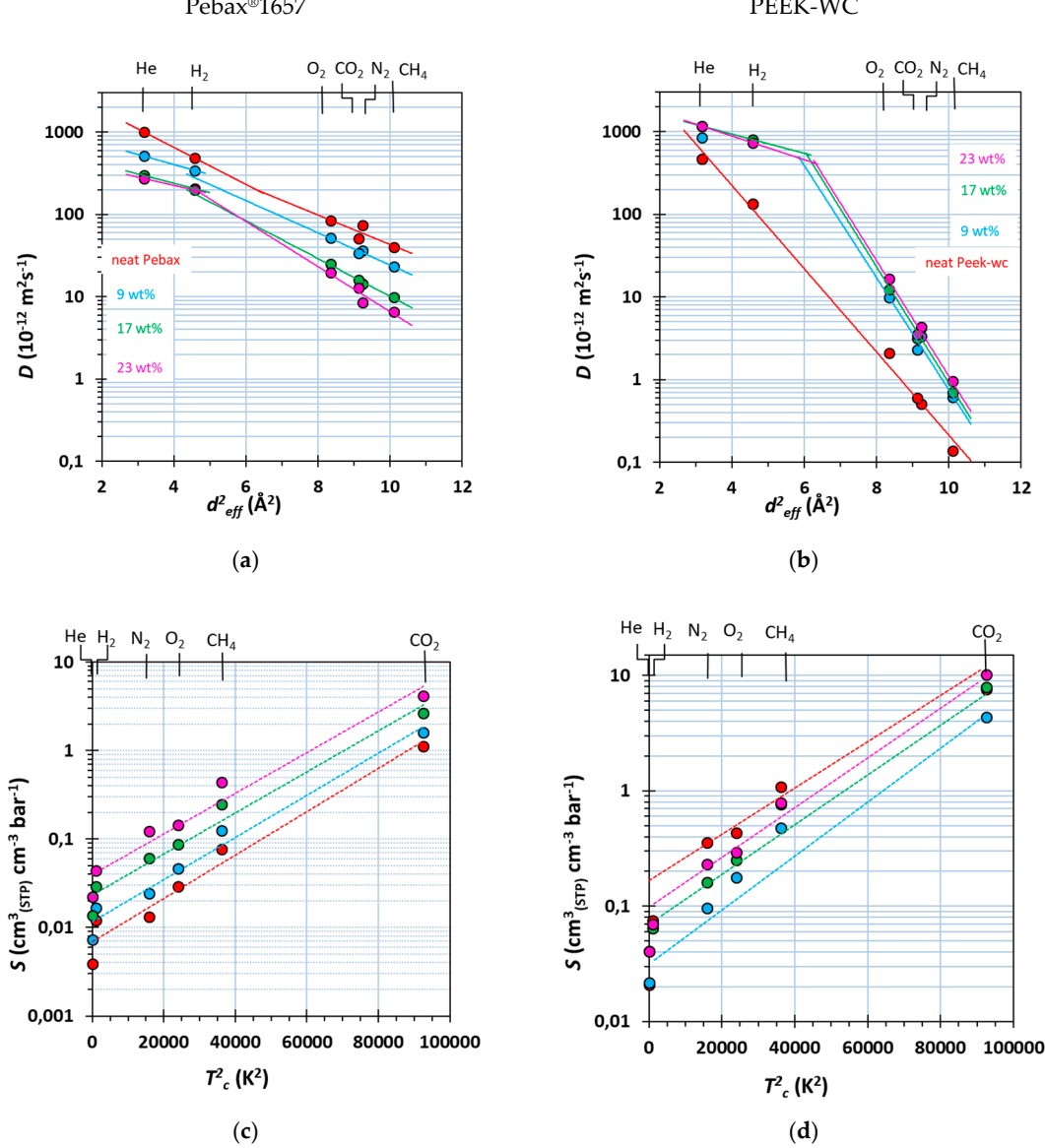

**Figure 7.** Correlation of the effective diffusion coefficient as a function of the molecular diameter of six light gases in (**a**) Pebax®1657/CuNi MMMs and (**b**) PEEK-WC/CuNi MMMs. Correlation of the effective solubility coefficient of the gases as a function of their critical temperature in (**c**) Pebax®1657/CuNi MMMs and (**d**) PEEK-WC/CuNi MMMs. The legend is identical in all individual graphs.

Contrary to the glassy PEEK-WC, the rubbery Pebax®1657/CuNi-MOF (23 wt %) membrane shows essentially pressure-independent permeability and selectivity (Figure 8a), after only a slight increase in permeability with pressure in the first run, apparently due to a certain conditioning of the sample. The transport properties of the Pebax-based membrane are dominated by the rubbery phase, and the dispersed MOFs only provide a higher gas sorption capacity in the membrane but do not significantly affect the overall performance.

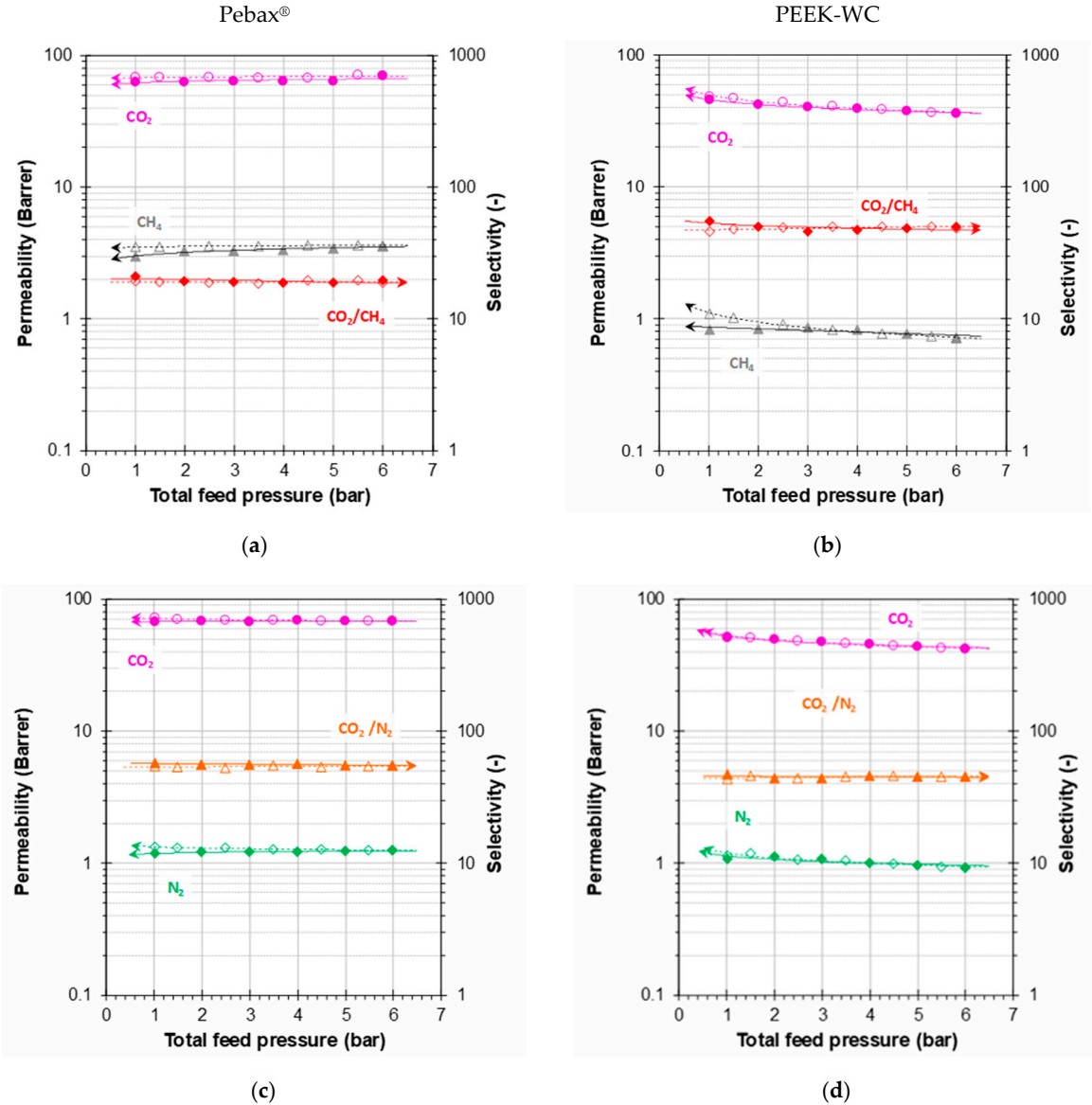

**Figure 8.** Pressure dependence of $CO_2$ and $CH_4$ permeabilities and $CO_2/CH_4$ selectivity using the binary mixture of $CO_2/CH_4$ (35:65 vol %) for Pebax®1657_23 wt % CuNi-MOF (**a**) and PEEK-WC_23% CuNi-MOF (**b**). Pressure dependence of $CO_2$ and $N_2$ permeabilities and $CO_2/N_2$ selectivity in binary mixture conditions for $CO_2/N_2$ (15:85 vol %) of Pebax®1657_23 wt % CuNi (**c**) and PEEK-WC_23% CuNi (**d**). Closed symbols for stepwise increase of the pressure and open symbols for the subsequent stepwise decrease of the pressure. Trend lines are plotted as a guide to the eye.

*3.5. Robeson's Plots and Performance Overview*

The gas transport properties for all membranes are summarized in the Robeson diagrams in Figure 9 for four gas pairs representing industrially important separations: $CO_2/N_2$, $CO_2/CH_4$, $O_2/N_2$ and $H_2/N_2$.

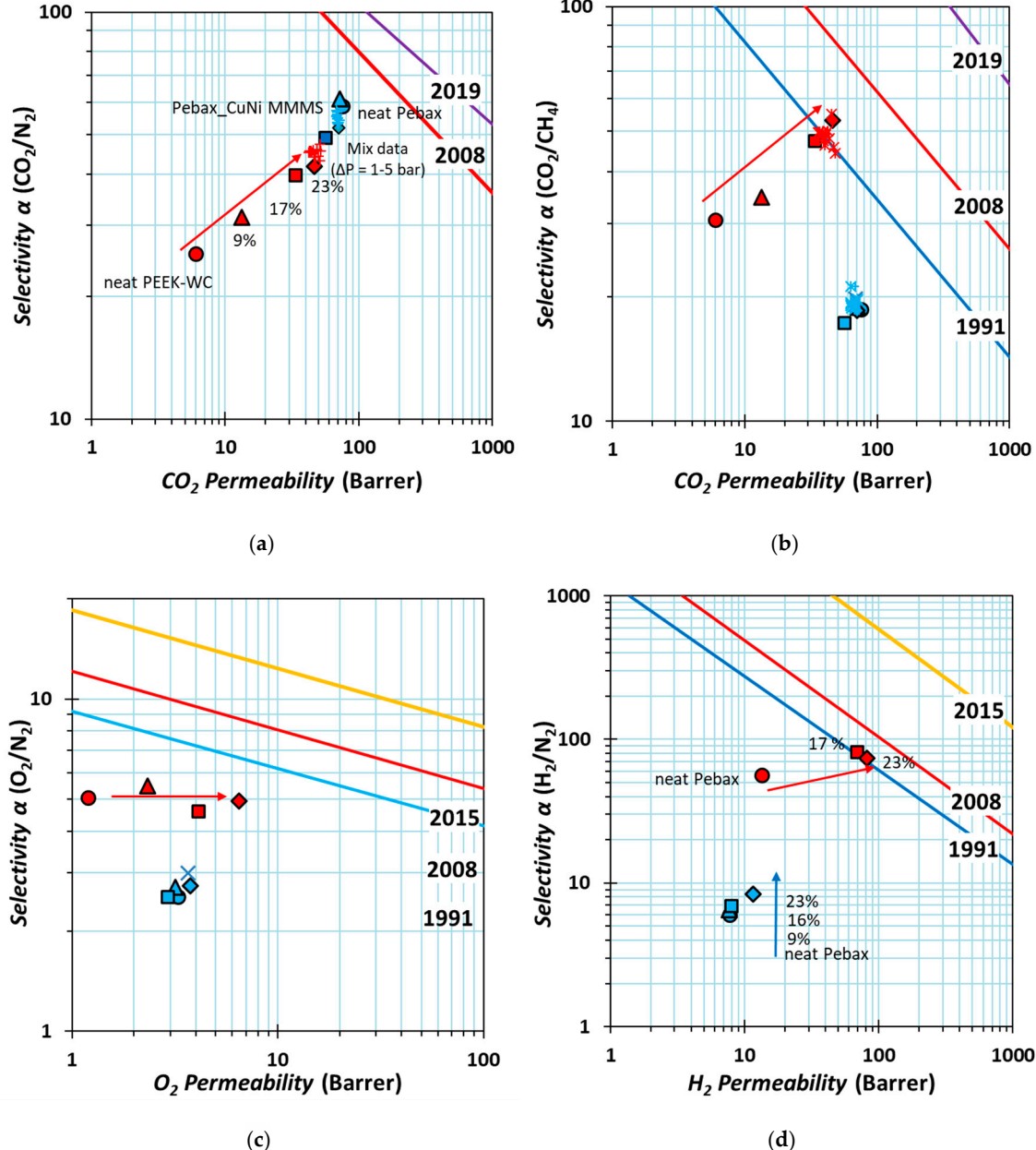

**Figure 9.** Robeson's plots for $CO_2/N_2$ (**a**) $CO_2/CH_4$ (**b**) $O_2/N_2$ (**c**) $H_2/N_2$ (**d**) showing the data of Pebax1657/CuNi (blue) and PEEK-WC/CuNi at different MOF loadings of 0 wt % (= neat polymers, •,•), 9 wt % (▲,▲), 17 wt % (■,■) and 23 wt % (♦,♦). Blue line 1991 upper bound, red line 2008 upper bound [8,9]; yellow line 2015 [60]; purple line 2019 upper bound [61]. The arrows qualitatively indicate the direction of increasing MOF loading. The clouds of blue "X" and red "X" symbols represent the mixed gas data in the pressure range of 1–6 bar for Pebax1657/CuNi-MOF at 29 wt % and PEEK-WC/CuNi-MOF at 23 wt % of MOF loading. Please refer to the electronic version for colour figures.

For all gases except $H_2$, the neat rubbery Pebax®1657 is more permeable than the neat glassy PEEK-WC. On the other hand, PEEK-WC is more selective than Pebax®1657 for all gas pairs, except for the $CO_2/N_2$ separation (Figure 9a). For the $CO_2/N_2$ gas pair, with the nearly identical diameters of the molecules, the selectivity is almost entirely due to solubility selectivity, which is higher in Pebax then in PEEK-WC. However, whereas CuNi-MOF in Pebax®1657 has a marginal effect, CuNi-MOF in the glassy PEEK-WC increases drastically not only the $CO_2/N_2$ selectivity (from 25 to 37), but also the

$CO_2$ permeability (from 6 to ~50 Barrer), near the $CO_2$ permeability of the rubbery Pebax®1657 (PCO$_2$ ~69 Barrer) (Figure 9a).

For $CO_2/CH_4$ separation, the PEEK-WC-based MMMs show far superior properties than the Pebax®-based membranes, which hardly change with the MOF concentration. A strong contribution of size selectivity between $CO_2$ and the much bulkier $CH_4$ molecules (Figure 7b) and a further increase of both solubility and diffusivity with increasing MOF concentration leads to enhanced selectivity (from 30 to ~50, above that of Pebax) and $CO_2$ permeability (from 6 to ~50 Barrer), exceeding the 1991 Robeson upper bound. For both mixtures, the performance is very similar to the ideal selectivity and pure gas permeability, indicating the absence of a significant coupling effect (Figure 9b).

The $O_2/N_2$ separation (Figure 9c) is mainly due to diffusion selectivity. Neither the permeability nor the selectivity changes much in Pebax®1657 upon the addition of CuNi-MOF, but in PEEK-WC the permeability of both gases increases 5-fold in the presence of 23 wt % CuNi-MOF, maintaining an almost constant $O_2/N_2$ selectivity.

Finally, the trend for $H_2/N_2$ separation is similar to that of $O_2/N_2$, but in this case, for both polymers, the MOFs also have a slightly higher $H_2/N_2$ selectivity than the neat polymers, as a result of increased diffusion selectivity. The $H_2$ permeability in Pebax® 1657 slightly increases from 8 to 11 Barrer, and the $H_2/N_2$ selectivity and permeability in the PEEK-WC/CuNi-MOF MMMs both strongly increase, surpassing the 1991 upper bound.

## 4. Conclusions

The development of novel, well-performing gas separation membranes requires a good understanding of their transport properties. This is even more important for complex systems such as mixed matrix membranes in which a porous filler is dispersed in the polymer matrix. Therefore, this work describes the comparison of the performance of two sets of membranes, glassy PEEK-WC membranes and rubbery Pebax®1657 membranes, with different concentrations of an oxamate-based CuNi-MOF. An increase of the Young's modulus for both membrane sets confirms that the MOF increases the stiffness of the polymer. In the case of the Pebax®1657 membrane, this is also accompanied by a slight shift of the melting peak of the PEO phase to higher temperatures, which suggests a good interaction of the MOF with the PEO phase. On the other hand, the presence of the MOF reduces the melting enthalpy and thus the overall crystallinity of both the PEO phase and the PA phase of Pebax®, whereas it does not affect the glass transition temperature of PEEK-WC. Only for PEEK-WC, along with the Young's modulus, the tensile strength also increases, but for both polymers, the maximum deformation decreases with the addition of the MOF.

In terms of transport properties, the permeability of PEEK-WC strongly increases with increasing MOF content, mostly due to an increase in the effective diffusion coefficient, whereas unexpectedly, the effective diffusion coefficient in Pebax®1657 drastically decreases upon addition of the MOF. This is almost completely compensated by an increase in solubility so that the permeability remains nearly constant. The increase in diffusivity favors especially the smaller gas species, and as a result, the PEEK-WC MMMs show a simultaneous increase in $CO_2$ permeability and $CO_2/CH_4$ selectivity, and the membranes with the highest MOF loading approach Robeson's 2008 upper bounds for these gas pairs. In the range of 1–6 bar(a), the mixed gas permeation tests with $CO_2/N_2$ 15/85 vol % and $CO_2/CH_4$ 35/65 vol % mixtures show only a weak pressure-dependence for the $CO_2/CH_4$ mixture in PEEK-WC with 30% MOF, typical for materials with dual-mode sorption behaviour, but not for the $CO_2/N_2$ mixture in PEEK-WC and for both mixtures in the Pebax-based MMMs.

In summary, the detailed analysis of the gas transport properties of the two series of MMMs highlights the enormous impact of the polymer matrix on the effectiveness of the same MOF when it is embedded in a rubbery or a glassy polymer. Even if the MOF improves significantly the gas solubility, this may not have a positive effect on the permeability if the diffusivity is not increased simultaneously. The successful development of better-performing MMMs can therefore not rely on studies of only the overall permeability but necessitates knowledge of the individual solubility and diffusion coefficient.

For both parameters, the values of the polymer and the porous filler should be carefully matched to yield optimum performance.

**Supplementary Materials:** The following are available online at http://www.mdpi.com/2076-3417/10/4/1310/s1, Table SI-1. List of MMMs prepared. Figure S-1 EDX of Pebax®1657/CuNi-MOF (a) and PEEK-WC/CuNi-MOF (b) at an accelerating voltage of 15 Kv. Figure S-2 SEM images of cross section for MMMs of Pebax®1657/CuNi-MOF and PEEK-WC/CuNi-MOF. Table S-2 Pure gas permeability, solubility and diffusion coefficients, and respective selectivity for neat Pebax1657 and Pebax1657/CuNi MMMs. Table S-3 Pure gas permeability, solubility and diffusion coefficients, and respective selectivity for neat PEEK-WC and PEEK-WC/CuNi MMMs. Table S-4 Mixed gas permeabilities and selectivities of PEEK-WC/CuNi 23 wt % membrane using binary mixture CO2/CH4 (35/65) at pressure of 1–6 bar. Table S-5 Mixed gas permeabilities and selectivities of PEEK-WC/CuNi 23 wt % membrane using binary mixture CO2/N2 (15/85) at pressure of 1–6 bar.

**Author Contributions:** E.E. and R.B. conceived, designed and performed the membrane preparation and characterization experiments under the supervision of J.C.J. and D.A.; J.F.S. and E.P. synthetized and provided CuNi-MOF; E.E. and A.F. performed the single gas permeation and DSC experiments; E.E. and J.C.J. performed mechanical tests. M.M. performed the mixed gas permeation experiments under the supervision of J.C.J. All authors analyzed the data and wrote the paper with similar effort. All authors have read and agreed to the published version of the manuscript.

**Funding:** This work was supported by Ministero dell'Istruzione, dell'Università e della Ricerca (Italy). Phenom-World B.V., Eindhoven (NL), is gratefully acknowledged for providing a Phenom Pro X desktop SEM for evaluation. C. Cantoni (Arkema Italy) is gratefully acknowledged for providing the Pebax®1657 pellets. E.P. acknowledges the financial support of the European Research Council under the European Union's Horizon 2020 research and innovation programme/ERC Grant Agreement No 814804, MOF-reactors.

**Acknowledgments:** Phenom-World B.V., Eindhoven (NL), is gratefully acknowledged for providing a Phenom Pro X desktop SEM for evaluation. R.B. thanks the MIUR (project PON R & I FSE-FESR 2014–2020) for predoctoral grants.

**Conflicts of Interest:** The authors declare no conflict of interest.

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
