# Peer review of "Glassy PEEK-WC vs. Rubbery Pebax®1657 Polymers: Effect on the Gas Transport in CuNi-MOF Based Mixed Matrix Membranes"

_applsci, doi:10.3390/app10041310_

Round 1

Reviewer 1 Report

Overall:

This report the synthesis of Cu-MOF and the incorporation in Pebax@1657 to make a mixed matrix membrane (MMM). This paper is interesting and promising, and the physical chemistry properties were well characterized. Although there have been many reports dealing with the similar MMM materials, the report here is promising in the future applications. I recommend accepting it for publication after minor revision. My comments are as follows.

Recently, many kinds of organic/inorganic composite membranes have been widely reported so it would be better that the authors can mention that there are other useful materials for adsorption applications. Several recent reports are recommended to be included in the Introduction part. Lin group: MOF/chitosan MMM. Journal of the Taiwan Institute of Chemical Engineers. 2018, 83, 143-151. Tung group: ZIF/PVA MMM. Angewandte Chemie International Edition. 2016, 55, 12793-12796. Chen group: MoS2/Pebax-1657 MMM. Journal of Membrane Science. 2019, 582, 358-366. How to check the crosslinking rate in this study? Is MOF crosslinked with the organic polymer by any chemical bonding?           Can the authors re-use the membrane? If so, please show the recycle test. Is it possible to maximize the amount of Cu-MOF to the membrane? If so, the authors can compare the effect of concentration on the separation performance. How thermal stability can this MMM achieve? It is well known that the CO2/CH4 is usually done at relatively high temperature.

Reviewer 2 Report

The present submitted paper deals with the investigation by several techniques (FT-IR spectroscopy, SEM and EDX analysis, tensile tests and DSC) and with the comparison of the gas transport performance of two sets of mixed matrix polymeric membranes: glassy PEEK-WC 396 membranes and the rubbery Pebax®1657 ones, with different concentrations of an oxamate-based 397 CuNi-MOF.

I find that the scientific problem presented in this paper is original and of scientific interest and the reported experimental and interpretative study is huge and complete.

Therefore, I have not major criticisms about this article.

I ask the authors to check some typing errors and misprints present along all the paper and also some English grammar and syntax.

Moreover the “Structural characterization” and the “Thermal and mechanical characterization” subsections may be written more accurately.
